# Toxic and Teratogenic Effects of Prenatal Alcohol Exposure on Fetal Development, Adolescence, and Adulthood

**DOI:** 10.3390/ijms22168785

**Published:** 2021-08-16

**Authors:** Dae D. Chung, Marisa R. Pinson, Lokeshwar S. Bhenderu, Michael S. Lai, Rhea A. Patel, Rajesh C. Miranda

**Affiliations:** Department of Neuroscience and Experimental Therapeutics, Texas A&M University Health Science Center, Texas A&M University, Bryan, TX 77807, USA; daehyukchung@tamu.edu (D.D.C.); marisa_pinson@tamu.edu (M.R.P.); lbhenderu@tamu.edu (L.S.B.); michaellai0420@tamu.edu (M.S.L.); rheapatel@tamu.edu (R.A.P.)

**Keywords:** prenatal alcohol exposure, alcohol, early childhood adversity, extracellular vesicles, developmental origin of health and disease, epigenetic modification, miRNA

## Abstract

Prenatal alcohol exposure (PAE) can have immediate and long-lasting toxic and teratogenic effects on an individual’s development and health. As a toxicant, alcohol can lead to a variety of physical and neurological anomalies in the fetus that can lead to behavioral and other impairments which may last a lifetime. Recent studies have focused on identifying mechanisms that mediate the immediate teratogenic effects of alcohol on fetal development and mechanisms that facilitate the persistent toxic effects of alcohol on health and predisposition to disease later in life. This review focuses on the contribution of epigenetic modifications and intercellular transporters like extracellular vesicles to the toxicity of PAE and to immediate and long-term consequences on an individual’s health and risk of disease.

## 1. Introduction

Alcohol exposure during pregnancy can have direct toxic and teratogenic effects on a fetus, because of its ability to pass directly and rapidly through the placenta into fetal organs including the developing brain [1]. Developmental exposure can result in fetal alcohol spectrum disorders (FASDs), an umbrella term covering the detrimental effects, including impairments on physical, neurological, and behavioral development, following prenatal alcohol exposure (PAE) [2]. These adverse effects can vary from one affected individual to the next and can include a range of structural and functional abnormalities, growth retardation, central nervous system dysfunction, and craniofacial dysmorphology [3]. Impairments have been documented in a variety of behavioral domains, including attention, self-regulation, and social judgment, as well as academic performance [4,5]. PAE’s effects on an individual are viewed both as toxic and teratogenic, due to its ability to have immediate toxic effects, i.e., cell death by oxidative stress or mitochondrial damage, but also have persisting teratogenic effects throughout development, i.e., changing and interfering with cell cycle and cell proliferation [6,7,8,9].

An estimated 1.1–5% of first-graders in 4 different US regions were identified as having FASDs, with prevalence attributed to unintended pregnancies and lack of knowledge on the harms of consuming alcohol during pregnancy [10,11]. A 2011 study reported that 45% of pregnancies in the US were unintended, with 1 in 10 pregnant women between the age of 18 and 44 in the US reporting alcohol consumption in the past 30 days and 1 in 33 reporting binge drinking [12,13]. FASDs are difficult to diagnose, a factor that has led to an underestimation on the number of individuals impacted by FASDs. Impediments to diagnosis include difficulty in acquiring an accurate and reliable history of maternal alcohol exposure during pregnancy, due either to an absence of the birth mother at the time of assessment or unreliable and underreported maternal self-reporting of alcohol consumption [14]. Another challenge, as documented by surveys, lies in knowledge deficiencies among physicians about even the cardinal diagnostic features of FASDs [15,16]. Moreover, neurobehavioral deficits associated with FASDs are predominantly likely to occur in the absence of identifiable craniofacial and growth characteristics, since facial development is only affected during a short gestational period in the first trimester, while brain development is susceptible to alcohol throughout pregnancy [14,17,18]. In addition, failure to properly diagnose results in missed opportunities for primary prevention, increased risk for secondary disabilities, and inappropriate patient care, which are all reasons that contribute to FASDs being the leading cause of intellectual disability [19].

While considering various ways PAE can have acute and progressive effects on an individual with FASDs, this review will focus on the mechanisms that allow toxic and teratogenic effects of PAE during the duration of an individual’s lifetime.

## 2. Candidate Mechanisms for Immediate and Persistent Effect of PAE

Prenatal alcohol exposure can have various structural and functional effects on a developing fetus. The severity of these effects, not only depends on the timing, duration, pattern, and dose of exposure, but can also vary across cell types and tissues and stages of fetal development. For example, when it comes to fetal neural stem cells, the teratogenic effects of PAE that lead to FASDs may be caused by disruption of cell proliferation, growth, differentiation, migration, and maturation [20]. A possible mechanism to explain such disruption is alcohol’s interference with growth factors that regulate cell division in the cerebral cortex, particularly IGF-I and IGF-II, preventing the normal growth and development of these cells and hindering fetal brain development [21]. Another example is neurons, where alcohol may cause defects in cell adhesion molecules, specifically L1, preventing neurons to clump together, which is needed to establish cell-to-cell contact for normal growth and development, and result in abnormal brain development [22,23,24].

Due to such complexity, there is no single global mechanism that explains PAE’s detrimental effects on fetal development and on lifelong consequences. It is likely that various mechanisms play a role at different times to affect multiple components of fetal development following PAE. There are candidate mechanisms that have been identified and researched on their respective contributions to prenatal alcohol-induced fetal damages. In order to understand the immediate and lifelong damaging effects of PAE, this review focuses on potential mechanisms that explain how prenatal alcohol can be a toxicant and a teratogen to an individual.

### 2.1. Epigenetic Mechanisms

The lifelong consequences of PAE may be mediated in part via mechanisms of epigenetic modifications, such as DNA methylation and histone methylation. DNA methylation at CpG islands is catalyzed by enzymes called DNA methyltransferases (DNMTs) and results in the silencing of genes [25]. Histone methylation can either be gene-repressive or gene-activating, dependent on the specific histone residue and precise modification, and is catalyzed by euchromatin histone methyltransferases (EHMTs) [26,27]. Ethanol is known to perturb one-carbon metabolism, an important source for the methyl group required for DNA and histone methylation [28,29]. PAE also perturbs the expression of associated methyltransferases, further impacting methylation status and downstream gene expression [30,31,32,33,34,35]. Specifically, in Japanese rice fish, continuous ethanol exposure (300 mM ethanol from 0 to 6 day post fertilization) during early embryogenesis disrupted the normal developmental rhythm of DNMTs and methyl binding proteins, which are crucial for recruiting chromatin modifying complexes [36]. Early disruptions like this are found in various organs and brain regions. In particular, PAE was demonstrated to alter developmental methylation rhythms of the hippocampus, resulting in slowed hippocampal formation and decreased proliferation and maturation of hippocampal cells [37], potentially explaining the contribution of PAE to impaired hippocampal-dependent learning and memory [38,39,40]. Moreover, ethanol consumption leads to elevated levels of acetate in circulation, a byproduct of ethanol as it is broken down by the liver [41]. Once metabolized to acetyl-CoA, elevated acetate levels result in increased histone acetylation levels, decreasing the interaction of N-termini of histones with phosphate groups of DNA and relaxing the structure between histones, thus making DNA more accessible for gene upregulation [42,43]. There is also an increase in histone acetyltransferase (HAT) activity while no change in histone deacetyltransferase (HDAC) activity as a result of PAE [44,45,46,47]. Altogether, PAE dysregulates the normal epigenetic programming by various mechanisms, and these teratogenic consequences have been shown to persist through the lifespan of the individual exposed in utero, manifesting later in life in health issues such as a predisposition for cancer and complications of the hypothalamic–pituitary–adrenal (HPA) axis [48,49,50,51,52].

### 2.2. miRNAs and Translational Regulation

MicroRNAs (miRNAs) are a class of small non-protein-coding RNAs, known to repress protein translation, that are alcohol-sensitive and can regulate the effects of alcohol on fetal brain development in animal models [53,54,55,56]. Because miRNAs are key players in stem cell self-renewal, cell cycle regulation, apoptosis, and development, dysregulation of miRNAs by PAE can have significant toxic and teratogenic effects on a developing fetus [57,58,59,60]. Understanding the interaction between PAE and miRNAs, and the regulatory role ethanol-sensitive miRNAs can have on an individual, may help us to better understand the ethanol-associated toxicology and teratology.

Recent studies have documented the effects of ethanol on miRNAs and the role miRNAs play to mediate ethanol’s teratogenic effects on a developing fetus. Ex vivo mouse model of isolated tissue from the fetal cerebral cortical neuroepithelium of second-trimester fetuses observed that ethanol (70 mM ethanol for 5 days) repressed the expression of four miRNAs (miR-9, miR-21, miR-153, and miR-335) in fetal neural stem cells (NSCs) and neural progenitor cells (NPCs) [53]. By repressing these miRNAs, ethanol induces the expression of the miRNAs’ target mRNAs, which is expected to result in increased translation of proteins, like Jag1 and ELAVL2/HuB in this case. Jag1 promotes the rapid proliferation of NSCs to establish a neuronal identity while ELAVL2/HuB promotes neuronal differentiation [61,62]. These data suggest that suppression of these ethanol-sensitive miRNAs may promote premature neuronal differentiation and deplete NSC/NPC population. Another study using a zebrafish model showed that ethanol (0–300 mM from 4 to 24 h postfertilization) suppresses miR-9 and miR-153 expression in embryonic development, resulting in significantly increased locomotor activity in juvenile zebrafish, resembling increased hyperactivity observed in children with FASDs [54]. These studies add to the cohort of studies that demonstrate the behavioral and anatomical consequences of miRNA dysregulation.

In addition, because circulating miRNAs secreted by cells into biofluids can be stable and potentially a cost-effective option as PAE biomarkers, a recent study assessed circulating miRNAs of human infant plasma from a heavily alcohol-exposed Cape Town Longitudinal Cohort. This study identified PAE-sensitive circulating miRNAs whose mRNA targets were enriched for pathways involved in PAE-induced growth deficits and infant recognition memory [63]. Another study assessed EV-associated miRNAs from sera of alcohol-fed mice (5% *v*/*v* ethanol with 36% ethanol-derived calories for 4 weeks) and plasma of alcoholic hepatitis (AH) patients and found seven inflammatory miRNAs that were upregulated in sera of chronic alcohol-fed mice compared to controls with both miRNA-192 and miRNA-30a being upregulated in alcohol-fed mice and AH subjects, providing potential diagnostic value as biomarkers in alcoholic hepatitis [64]. Future studies should focus on alcohol-sensitive miRNAs to help identify and diagnose PAE infants that may exhibit FASD-related deficits in growth and cognition.

### 2.3. Cytokine and Immune Response

Cytokines, generally classified as pro-inflammatory or anti-inflammatory depending on how they affect the immune system, are small proteins produced by many cell types and can affect cell proliferation, migration, and death. Ethanol has been known to regulate various cytokines’ activity that is dependent on cell type, ethanol dose and duration, and the co-presence and characteristics of pathogens. Clinical and animal studies have demonstrated pro-inflammatory cytokine imbalances and immune disturbances in PAE offspring [65,66,67]. While some pro-inflammatory cytokines are developmentally regulated and normally expressed at significant levels in the developing brain even without any inflammatory event, altered levels of these cytokines due to PAE can have detrimental effects on brain development to result in long-term consequences on an individual [68].

In PAE animal models, moderate to high ethanol exposure resulted in increased pro-inflammatory cytokine production and heightened neuroimmune activation in the neonatal rat brain, leading to ethanol-induced neurodegeneration in a developing brain [65,66,67]. On postnatal day 8, female PAE (6.37% *v*/*v* ethanol with 36% ethanol-derived calories from gestational day 1 to 21) rat pups’ brain demonstrated significantly increased levels of cytokine interleukin 5 (IL-5) and interleukin 6 (IL-6) in the prefrontal cortex and tumor necrosis factor alpha (TNF-α), interferon gamma (IFN-γ), IL-1β, IL-2, IL-4, and IL-5 in the hippocampus, compared to controls [66]. In the same study, levels of cytokine IL-2, IL-1β, and TNF-α in the hypothalamus of the PAE group on postnatal day 8 were lower compared to controls. Increased levels of TNF-α in a developing brain can be neurotoxic by inducing oligodendrocyte apoptosis and myelination impairments in vitro and disturbing the blood–brain barrier in vivo [69,70]. Likewise, IL-1β and IL-2 overexpression in vitro has shown oligodendrocyte toxicity and apoptosis induction in brain [71,72].

Finally, the inflammatory effects of PAE may synergize with the effects of other concurrent life experiences. For example, one study used a combined rodent model of PAE (ad libitum liquid ethanol diet with gestational day 1–34%, 2–66%, 3 to 21–100% ethanol diet) with a naturalistic model of early-life adversity that deprived rat mothers of sufficient bedding from P8-12 [73]. When evaluating PAE animals’ peripheral and central immune responses to early-life adversity at P12, serum (peripheral) C-reactive protein levels were highest for PAE pups exposed to adversity, with increased IL-10 and reduced KC/GRO levels in the amygdala (central), while early-life adversity reduced serum (peripheral) TNF-α, KC/GRO, and IL-10 levels in control pups compared to PAE pups.

Due to the key roles that cytokines play during brain development, altered cytokine balance and neuroimmune dysregulation observed in PAE offspring during critical early development may result in a permanent change in a developing brain, providing a possible insight into the high rate of mental disorders in children with FASDs and increased susceptibility to aging mental health disorders observed in adults with FASDs. Future studies looking into cytokine and immune-focused therapeutic intervention during specific early-life stages could be a novel approach to mitigate some of the toxic and teratogenic effects of PAE on individuals.

### 2.4. EVs and Intercellular Transfer of Toxicity

In an effort to mitigate the effects of PAE, extracellular vesicles (EVs) have been of interest due to their involvement in intercellular communication, biocompatibility, low toxicity, and non-immunogenic properties [5,74,75]. EVs are membrane-bound vesicles typically ranging from 50–500 nm that carry proteins, lipids, and nucleic acids between cells and be crucial mediators of cell-to-cell communication and disease pathogenesis (Figure 1) [76,77]. Based on the size, mechanism of biogenesis, and biological content, EVs can be categorized into exosomes, microvesicles, and apoptotic bodies [78]. Exosomes are produced from multivesicular bodies, which contain intraluminal vesicles, through endosomal pathways that include internal budding and exocytosis [79,80]. Microvesicles, on the contrary, are produced through outward budding of the plasma membrane, while apoptotic bodies result from cell fragmentation during programmed cell death [81].

EVs carry a diverse group of molecules that are determined by the EV pathway and environment of their parent cell [82]. Endosomal sorting complex required for transport (ESCRT) proteins, Alix, and TSG101 are standard markers of exosomes that derive from any parent cell since they are necessary for the biogenesis of multivesicular bodies. Integrins and tetraspanins, such as CD63, are common membrane proteins found in exosomes along with cytosolic proteins, such as Hsp70 and Hsp90 [83].

EVs may be playing significant roles in regulating immune response and tumor diagnosis as nanovesicles for therapeutic agent delivery and special biomarkers, respectively [74,84]. For example, EVs derived from immunosuppressive dendritic cells show a significant effect on animal inflammation and autoimmune diseases [85]. Additionally, EVs have been found useful as cancer biomarkers due to their availability in biological fluids, high stability, and capability of representing parent cells and their associated roles in critical physiological activities [84]. An in vitro study of the plasma of patients newly diagnosed with colorectal cancer found miR-125a-3p in EVs to be an effective diagnostic biomarker of early-stage colon cancer [86]. For in vivo environments of mice exposed to ethanol, EVs have shown to vary cargo composition compared to control environments due to their regulated mechanism of cargo sorting [79]. Compared to control groups, binge alcohol consumption results in EVs with elevated levels of miRNA-122, a liver-specific miRNA, and decreased levels of miR-29b, a tumor suppressor miRNA [87].

It has been shown that environmental stressors alter the secretion and composition of EVs, therefore controlling their function. In neurogenesis, EVs coordinate behavior between various cell types of the developing brain [88]. During the neonatal period of mice, in vivo neural stem cell (NSC)-derived EVs cause a decrease in microglial complexity and an increase in cytokine production due to preferential signaling of microglia [89]. The teratogenic effects of ethanol on the cortex result in alterations of EV signaling pathways [90]. An in vitro mouse study found that the decrease in miR-21 following ethanol exposure could be the result of an adaptive response to restore neurite outgrowth and decrease inflammation [53,90]. Following PAE, an increase in pro-inflammatory microglia in the cortex is found [91]. This increase seems to be triggered in part by EV signaling from developing cortical cells and involvement of TLR4 receptors as mediators of the microglial inflammatory response following PAE [90]. Another study observed that ethanol exposure (70 mM ethanol concentration for 3 days) on ex vivo murine fetal neural stem cells significantly elevates a group of ethanol-sensitive miRNAs in NSC-derived EVs [5]. Overexpression of miR-140-3p, one of the elevated EV miRNAs, in naïve fetal neural stem cells results in a significant increase in the proportion of S-phase cells and a decrease proportion of the G_0_/G_1_ phase compared to the controls. miR-140-3p overexpression results in abnormal neuronal progenitor cell growth and maturation caused by a predominance of astroglial maturation over neural and oligodendrocyte differentiation [5]. In an in vivo study done on macrophages of humans confirmed with alcoholic hepatitis, macrophage-derived EVs of humans with alcoholic hepatitis were introduced to naïve monocytes, which induced the monocytes into M2 macrophages through differentiation and polarization [92].

PAE decreases the proliferation of adult NSCs, possibly due to ethanol’s impact on EV miRNA cargo of peripheral mesenchymal cells that signal neural stem cells and progenitor cells to differentiate instead of proliferating [90,93]. The decreased adult NSC proliferation is possibly due to EV repair signaling mechanisms that need to be further investigated [94]. In an in vitro mouse study, there were deficits in adult neurogenesis following ethanol exposure which is likely due to intrinsic dysregulation of neural stem progenitor cells [93].

Studies on the mechanisms of EVs that contribute to ethanol’s dysregulation of neural stem progenitor cells need to be further elucidated. Based on the role of EVs on PAE and its connection to FASDs, future studies should focus on miRNAs that are loaded into EVs, due to their known role within ethanol teratogenesis and their potential role as a biomarker for FASDs, and EVs as an intercellular signaling mechanism to elicit an immune response in a PAE individual.

Finally, it should be acknowledged that the mechanisms that mediate the developmental effects of ethanol are likely to be dependent on the timing of exposure. Evidence for this possibility comes from studies on ethanol effects on pathways that promote cell survival and apoptosis. For instance, ethanol exposure between postnatal day (P) 4 and 7 (via vapor chamber with the mean concentration of 87.42 to 106.64 mg/dL) resulted in increased cerebellar levels of the pro-apoptotic protein Bad at P4 but a decrease in the proapoptotic pJNK and an increase in the protective pAkt kinase at P7 compared to controls [95]. In another study by the same group, neonatal rats exposed to ethanol during the early postnatal period (35–39% ethanol-derived calories from gestational day 1 to birth) exhibited significantly increased nerve growth factor (NGF) in postnatal day 1 cortex/striatum that returned to control levels by P10. In contrast, postnatal ethanol exposure from P4 to P10 (ethanol vapor chamber with the mean concentration of 87.42 to 106.64 mg/dL) elevated brain-derived neurotrophic factor (BDNF) in hippocampus and BDNF and NGF in cortex/striatum at P10 [96].

## 3. Effects of PAE beyond Child Development

Prenatal alcohol exposure has well-known toxic effects. Studies have shown that PAE not only affect the neurodevelopmental and craniofacial abnormalities seen in fetal alcohol syndrome (FAS), but also affect various organ systems including the cardiac, renal, endocrine, and reproductive systems [97]. The abnormalities associated with PAE begin during development and have the potential to persist throughout a person’s life and increase susceptibility to diseases, such as diabetes and hypertension [98].

### 3.1. Puberty

A critical period in human development that can affect a person’s lifelong health status and quality of life is puberty. Many profound maturational changes, including physical, gonadal, and neurodevelopmental changes occur during this time of a person’s life [99]. The effects of PAE on an individual can resurface and manifest during the pubertal period.

#### 3.1.1. Hypothalamic-Pituitary-Gonadal Axis and PAE

In humans, the onset of puberty begins with activation of the hypothalamic-pituitary-gonadal (HPG) axis, which leads to gonadotropin-releasing hormone (GnRH) release from the hypothalamus [100]. GnRH release leads to secretion of luteinizing hormone (LH) and follicle-stimulating hormone (FSH) from the anterior pituitary gland which goes on to activate the gonads and develop the clinical features of puberty and sexual maturity [99]. The HPG axis is first active during the embryonic and early postnatal period of life, and the axis is reactivated during the start of puberty (Figure 2) [99]. The exact mechanism by which the HPG axis is reactivated is unknown but has been shown to be partly influenced by genetic factors and environmental factors [100]. However, the timing of puberty has well-documented health consequences. Early puberty onset, clinically termed precocious puberty, has been associated with diseases such as metabolic syndrome, cardiovascular diseases, and a range of cancers [99,101]. On the other hand, late puberty onset has been associated with diseases including neuropsychiatric diseases, decreased sperm counts, and longer time-to-pregnancy [101,102]. One environmental factor that has been shown to alter the HPG axis and pubertal timing is PAE [100]. Since the onset of puberty has been associated with various health conditions and puberty is a critical period of human development, it is vital to understand the progressive effects of PAE on puberty and the HPG axis.

#### 3.1.2. Animal Studies on HPG Axis and PAE

In various animal studies, PAE had a significant impact on the HPG axis and has been shown to delay the onset of puberty [99,100]. A study with female rat offspring was documented either before puberty onset, around the time of puberty, or in young adulthood after PAE [100]. The study found that rats with PAE had a delay in sexual maturation, among various other differences when compared to control animals. Interestingly, this study showed that PAE rats had alterations in Kisspeptin 1 (Kiss1) gene expression [100]. Kiss1 is a gene that plays an important role in the regulation of the HPG axis and has been previously shown to coincide with the onset of puberty [100]. The alterations in Kiss1 gene expression by PAE could be one mechanism by which pubertal onset is affected in these animal studies. Similar results were shown in a study looking at the development of seminiferous tubules and the onset of spermatogenesis in a rat model [102]. Development of seminiferous tubules and spermatogenesis have been previously shown to correspond to puberty onset in males. The study found that the development of the seminiferous tubules and onset of spermatogenesis were delayed in PAE male offspring [102]. These studies and various others have shown that PAE does lead to delayed onset of puberty in animal models [103,104,105].

#### 3.1.3. Human Studies on HPG Axis and PAE

Studies in human populations on the effects of PAE on pubertal onset have yielded varied outcomes. A study of 265 adolescents, both male and female, found no changes in Tanner stages, which is a system to follow the development of secondary sexual characteristics, or the age of menarche between PAE adolescents and non-PAE adolescents [106]. Similarly, it was found that in a Danish cohort of 3169 females that PAE did not lead to an accelerated age of menarche, but that cigarette exposure alone during pregnancy led to an accelerated age of menarche [107]. This outcome highlights the possibility that prenatal exposure to toxins can affect the onset of puberty in humans. The paper did note that lack of association between PAE and accelerated age of menarche despite previous studies showing otherwise could be due to different exposure conditions. Another study assessed a marker of pubertal timing, the height difference in standard deviations (HD:SDS), which is the difference between pubertal height and adult height in standard deviations, and has been shown to correlate with the age of peak height velocity during puberty [108]. This study found a sex-dependent significant change where PAE led to a lower HD:SDS in boys, indicating an earlier onset of puberty, but not in girls [101]. This study also adjusted for childhood BMI since childhood obesity is associated with earlier puberty and found that the association between PAE and lower HD:SDS in boys was still present [101]. Another study compared 2522 sons of Danish women with weekly alcohol consumption (categorized into four groups: 0, 0.5–1.5, 2–4 or >4 drinks/week; one drink defined as 0.33 L of beer) during pregnancy to women with binge drinking episodes (eight or more alcoholic drinks on a single occasion) during pregnancy. The study used pubertal indicators such as first nocturnal emission and voice break to determine the onset of puberty [109]. They found no statistically significant delay in pubertal development among women with weekly alcohol consumption during pregnancy. However, they did find a delayed onset of puberty with pregnant women who had ≥5 binge drinking episodes since the boys had a delayed age for first nocturnal emission and voice break compared to unexposed boys [109]. This is an important finding because it shows that looking at the amount and pattern of alcohol consumption by pregnant women is vital to understanding the effects of PAE on the pubertal onset.

Overall, studies looking at how PAE affects pubertal onset in humans have been inconsistent when compared to animal studies which more consistently show that PAE leads to delayed onset in puberty. The inconsistencies in human studies have various potential causes and highlight the complicated nature of PAE pathogenesis and pubertal onset. Varied results in human studies could be potentially explained by looking at confounding environmental factors, such as nutrition, socioeconomic standing, and level of stress during pregnancy and during childhood. For example, one study showed that mothers who had children diagnosed with FASDs had a lower intake of several nutrients, such as calcium, which is important for bone development, and riboflavin, which plays an important role in vital biochemical reactions [110]. This lack of nutrients during the prenatal period also has the potential to affect pubertal onset and a person’s health throughout their life. The effects of PAE are multifactorial and more research is needed to elucidate the effects of PAE so interventions can be made to attenuate debilitating consequences. Even if PAE does not directly affect the onset of puberty, the effect of PAE on the HPG axis is still vital to understand since the reproductive system is intimately connected with the HPG axis. Various studies have shown that PAE leads to changes in both female and male reproductive health.

#### 3.1.4. Reproductive Health and PAE

Male reproductive health relies heavily on the physiological role of testosterone [111]. The elevated levels of LH and FSH during the beginning of puberty result in increased testosterone levels resulting in the onset of spermatogenesis and the development of secondary sexual characteristics including testicular growth, male genitalia, and pubic hair [111,112]. Control of testosterone through the HPG axis is also essential throughout the lives of males because testosterone is involved in the maintenance of various organs and directly influences male fertility [111]. Studies in humans have shown that PAE leads toward delayed pubertal development and lower semen quality and sperm concentration [112,113]. This is an important finding because poor semen quality is one of the main causes of fertility issues in males [112]. Interestingly, the effects of PAE on testosterone levels have been varied. Elevated testosterone levels in both females and males were found when looking at a sample of 265 African American adolescents [106]. This study found that salivary testosterone concentration was associated with higher average daily maternal alcohol use, but was not related to maternal cigarette, marijuana, or cocaine use during pregnancy [106]. This study also found that there were no changes to Tanner stages or first age of menarche, indicating no changes in the onset of puberty. Another study that looked at testosterone concentration in adults who had PAE found no association with PAE and levels of testosterone and several other reproductive hormones including LH and FSH [113].

Female reproductive health is especially influenced by the prenatal period because women develop their lifetime supply of gametes before birth. These gametes remain as non-growing primordial follicles until the start of puberty and menstruation [114]. This ovarian reserve declines throughout a woman’s lifetime and determines the age of menopause and lifespan of fertility. Studies have shown that environmental and nutritional exposures during the prenatal period affect ovarian reserve, and therefore affect a woman’s fertility and reproductive lifespan [114,115]. Studies focusing on the effects of PAE on female reproductive health have been mainly done in animal models. One study which used rats to study ‘special occasion’ drinking, which is a low exposure of alcohol during pregnancy and is relatively common, found that there was no association on markers regulating follicle numbers or other reproductive markers and PAE [114]. However, another study looking at higher levels of PAE found that there was a decrease in the number of oocytes in rats with PAE [115]. Furthermore, this study found that the level of ovarian insulin-like growth factor was increased, showing a possible mechanism behind accelerated early folliculogenesis and early depletion of primordial follicles [115].

There is still a substantial amount of research required to understand the toxic effects of PAE, especially when it relates to pubertal onset and reproductive health. The animal studies showing that PAE leads to a later pubertal onset indicate that PAE does play an intermediary role, but the inconsistency in human studies emphasizes the complicated nature of PAE pathogenesis and shows that the amount of exposure, the timing of exposure, and many other confounding factors are also involved. By focusing on candidate mechanisms, including miRNAs and EVs, that PAE uses to have immediate and persistent effects on an individual, this can help ease the complexity of PAE pathogenesis. Examining how PAE affects the HPG axis is not only vital to understanding its effects on the onset of puberty, but also on reproductive health which is heavily reliant on the HPG axis.

### 3.2. Prenatal Alcohol as a Contributor to Disease Burden in Adulthood

The fetal origins hypothesis, formulated by Dr. David J. Barker, postulated that metabolic and cardiovascular disease in adulthood could be elicited by the negative influences of external factors, such as undernutrition, in utero [116,117]. Barker’s hypothesis was later expanded to encompass studies on the persistent effects of developmental exposures to a variety of environmental agents on health outcomes and other diseases observed in adulthood. These studies spurred the formation of the developmental origins of health and disease (DOHaD) hypothesis which served as a theoretical framework to identify mechanisms that mediate the effects of developmental exposures on health outcomes and disease predisposition in later life [118]. This area of research has contributed towards the understanding of how insults in utero, such as prenatal alcohol, may result in toxic and teratogenic consequences that persist into adulthood [119,120,121,122]. One study reported that individuals diagnosed with FASDs have a higher mortality rate at every age range beginning in the late teenage and young adulthood [123]. This increase in mortality rate is contributable in part to systemic diseases, such as metabolic, cardiovascular, endocrine, renal, and respiratory disease, which are the second leading cause of death at 25% after mental health/drug overdoses at 27% [123,124,125,126,127]. A DOHaD approach to understanding this increase in mortality and systemic disease reveals the pathogenic underpinnings in individuals with FASDs and the long-term effects of the toxic and teratogenic mechanisms of PAE (Figure 3).

#### 3.2.1. Cancer

When normal epigenetic programming is perturbed during development by teratogens such as alcohol, this results in a potential increased risk for carcinogenesis where cancer-promoting pro-growth pathways are not silenced, when they normally would be, while stem cells mature and differentiate in organogenesis [128,129,130]. In particular, while PAE results in both DNA hyper- and hypomethylation, hypomethylation more often occurs in the gene body and promoter regions [131], suggesting less silencing of pathways of involved genes. This paradoxical mix of hyper- and hypomethylation is also seen in cancer, whereby chance results in silencing of tumor suppressor genes and upregulation of proliferation genes [132]. This seems to translate into higher rates of cancer observed in individuals with FASDs aged 18–44 compared to the general population (3.75% FASDs vs. 2.0% general population; 1.9 fold higher) [125]. PAE causes stable epigenetic marks on the genome that persist long after initial exposure [48,49,50,51,52,133,134], potentially predisposing specific tissues to tumorigenesis in adulthood since developmental and cancer biology are so closely linked [129,130]. This predisposition could be linked temporally to the stage of organogenesis and the time point during the pregnancy at which alcohol was consumed, making those organs that were developing during the exposure period the most susceptible to carcinogenesis later. This view helps to explain the increased susceptibility observed in PAE adult humans and animals to leukemia [135], mammary tumors [136], and prostate tumors [137], all of which have been linked to epigenetic origins [138,139,140]. PAE may be altering the epigenetic landscape in tissues such as these during development, thus interfering with the normal silencing of proliferation and cellular migration pathways as cells differentiate, resulting in a predisposition for tumorigenesis in childhood and adulthood.

#### 3.2.2. Hypothalamic-Pituitary-Adrenal (HPA) Axis

PAE has been shown to have teratogenic consequences on the HPA axis, altering epigenetic programming to decrease proopiomelanocortin (*Pomc*) expression in the hypothalamus [51,52]. *Pomc* is the precursor gene of β-endorphin, a neuropeptide that under normal conditions inhibits corticotrophin-releasing hormone (CRH) release from the hypothalamus [141,142]. Therefore, decreased levels of *Pomc* and decreased β-endorphin resulted in increased CRH release [143], leading to elevation of the HPA axis response to stressful stimuli in 1-month-old rats that were exposed to alcohol in utero [144]. HPA axis dysfunction in PAE is linked to many symptoms associated with FASDs, including deficient stress response, depression, anxiety, impaired immunity, and metabolic disorder [144,145,146,147]. For example, in depression, a hyper-responsive HPA axis and deficits in feedback regulation result in high basal levels of cortisol [148,149], and HPA disturbances and hypercortisolemia have been identified in 40–60% of depressed patients [150,151,152,153,154]. By contributing to an increased HPA drive, PAE may be priming an individual for developing depression later in life after exposure to additional trauma or environmental stressors that individuals with FASDs are at higher risk for, such as early maternal death, child abuse and neglect, living with a parent with chronic alcohol use disorders, removal from the home by authorities, and repetitive periods of foster care/transient home placements [19,155,156]. Altogether, this reveals how the lasting teratogenic consequences of PAE on the HPA axis during development through epigenetic modifications impair physiological processes in adulthood, impacting the quality of life and health of the individual.

#### 3.2.3. Neural Crest and the Emergence of Heart Disease and Immune System Dysfunction

Ethanol can act as a toxicant by inducing cell death during crucial windows of development, with cranial neural crest cells (NCCs) being particularly sensitive [122]. Ethanol mobilizes calcium through the IP3 pathway [157], and it is by this rapid rise in intracellular calcium (Ca_i_^2+^) that ethanol induces apoptosis in cranial NCCs [158]. Elevated Ca_i_^2+^ induces apoptosis in PAE by triggering the selective activation by phosphorylation of calmodulin-dependent protein kinase II (CaMKII) in cranial regions of the neural folds, sparing the more caudal NCCs [159]. Activated CaMKII, in turn, phosphorylates and destabilizes the transcriptional effector β-catenin [160,161]. During normal development, the Wnt/β-catenin signaling contributes to the endogenous death within rhombomeres 3 and 5 [162], which are rhombomeres that are particularly sensitive to PAE-induced apoptosis [163]. This overlap in effector pathways between developmentally programmed cell death and PAE-induced cell death explains why cranial NCCs contained within rhombomeres 3/5 are particularly sensitive to the toxicant consequences of ethanol. Moreover, while these cranial NCCs give rise to numerous important cell populations that contribute to facial morphology and cranial nerve development, the potential lifelong impact on health is because cranial NCCs give rise to cardiac NCCs, a subset of which become thymic mesenchymal cells [164]. These are cell populations from which FASD-associated diseases such as congenital heart disease and impaired immune function originate [146,165,166,167].

#### 3.2.4. Congenital Heart Disease

One toxicant effect of in utero exposure to ethanol is congenital heart disease (CHD). PAE results in an increased risk of the development of specific types of CHDs, such as conotruncal defects and transposition of the great arteries [165,166]. These types of CHDs are linked to cardiac NCC apoptosis and abnormal epigenetic modifications [168], both of which may result from the toxic and teratogenic consequences of PAE, respectively. Tight regulation of histone methylation and acetylation status is required during development [169], to prevent abnormal cardiogenesis [170]. Acetylation status of H3 histone lysine 9 (H3K9) is one link between PAE and CHD. Studies demonstrated that increased H3K9 acetylation after ethanol exposure resulted in increased expression of key cardiac development genes (*Gata4*, *Mef2c*, *Nkx2.5*, and *Tbx5*) [47,171] and in congenital heart defects in fetal mice similar to those associated with FASDs [44].

CHD patients are more likely to survive to adulthood as diagnostics and surgical interventions have improved for CHD [172]. The majority of patients with CHD seen in the clinic are adults, and they will likely continue to represent the largest proportion of patients requiring life-long medical care [173,174]. This is because even after surgery in childhood to fix congenital defects, adult CHD patients are at an increased rate of CVD risk factors (e.g., hypertension and obesity) [175,176,177] and also in general have an increased risk of CVD (e.g., myocardial infarction and stroke) [175,178,179,180,181,182]. Underlying CHD may in part contribute to the elevated risk of CVD observed in the population with FASDs [183]. Additionally, acute alcohol exposure during early development permanently decreases nephron number in rats, contributing to hypertension in adulthood [184]. Moreover, increased vascular resistance in cranially-directed blood flow has been shown to worsen stroke severity in adult mice who were prenatally exposed to ethanol [185]. Altogether, the toxicant effects of PAE on cardiac NCCs and nephrogenesis may establish a predisposition for CVD throughout an individual’s lifespan.

#### 3.2.5. Immune System

Another potential toxicant consequence of PAE is the increased apoptosis of cardiac NCCs that contribute to the development of the thymus. Lymphoid progenitors migrate to the thymus between gestational day 10–11 in mice where they undergo selection and maturation processes [186], resulting in the generation of diverse T cell populations essential for the establishment of cellular immunity [187]. Additionally, the thymus is comprised of blood vessels, connective tissue, and epithelial cells, forming a highly specialized microenvironment. Thymic deficits are associated with abnormal NCC survival and migration diseases, such as DiGeorge syndrome [188] and CHARGE (coloboma, heart defect, atresia choanae, retarded growth and development, genital hypoplasia, ear anomalies/deafness) syndrome [189,190]. Specifically, during normal development, cardiac NCCs migrating in the 3rd pharyngeal arch on gestational day 8 to form the thymus become the mesenchymal and perivascular cells and connective tissue of the thymus [191,192,193,194]. Studies that ablated cranial NCCs have shown that lack of NCC-derived mesenchyme in the thymus resulted in subsequent abnormal thymic development, revealing NCCs have an important role in thymus organogenesis [195,196]. One study did show that NCC-derived pericytes are critical for T cells to exit the thymus and enter circulation as part of the maturing immune system [192], though the detailed mechanisms of cell-to-cell interactions between NCC-derived cells and lymphocytes remain to be elucidated. Because PAE has been shown to cause cell death within cardiac NCC populations [122], this crucial population cannot migrate to the developing thymus to fulfill their normal role in organogenesis and T cell maturation and proliferation, resulting in the abnormal thymic development and reduced thymocyte number observed in PAE [167,197,198,199,200]. This explains at least in part the impaired immune function associated with PAE [146]. Because they cannot mature properly without crucial interaction with NCCs, T cells are unable to generate normal proportions of T cell subsets [199,201], resulting in T cell dysfunction [200,202,203]. Dysregulation of T cell populations and function contributes to increased susceptibility to infections across the lifespan [146,204,205] and other autoimmune/inflammatory-related diseases, such as adjuvant-induced arthritis (a model for rheumatoid arthritis) [206] and adult-onset neuropathic pain resulting from a predisposition for allodynia in individuals with FASDs [202]. Taken together, this possibly reveals that the underlying causes for lifelong impaired immunity in the population with FASDs can be traced back to the toxicant effects of PAE on cardiac NCCs migrating to the thymus.

The persistent effects of alcohol as a developmental exposure on health outcomes and disease predisposition in later life has been continuously studied and examined evermore extensively since the DOHaD hypothesis. Because individuals with FASDs have increased mortality rate that is contributable in part to systemic diseases, future studies need to continue this endeavor of revealing the pathogenic underpinnings of FASDs and the long-term effects of PAE mechanisms.

## 4. Conclusions

Alcohol is a known toxicant, causing cell death in a fetus, and a teratogen, altering cell cycle and function in a developing fetal brain, with PAE having immediate and persisting effects on an individual with FASDs. Due to the complexity and variety of mechanisms that PAE use to impact an individual with FASDs, this review of the literature focused on epigenetic modifications and intercellular mechanisms in immune response, miRNAs, and EVs that may be causing acute and life-long effects of PAE during development, adolescence, and adulthood. As revealed by our review of the literature, additional research is needed to elucidate how these mechanisms can mediate the long-term consequences of early life experiences. By understanding how prenatal alcohol exposure leads to acute and later effects of FASDs, it may be possible to better identify individuals with FASDs at an earlier stage, to allow for a faster and increased intervention period and mitigate the disease effects of PAE.

## Figures and Tables

**Figure 1 ijms-22-08785-f001:**
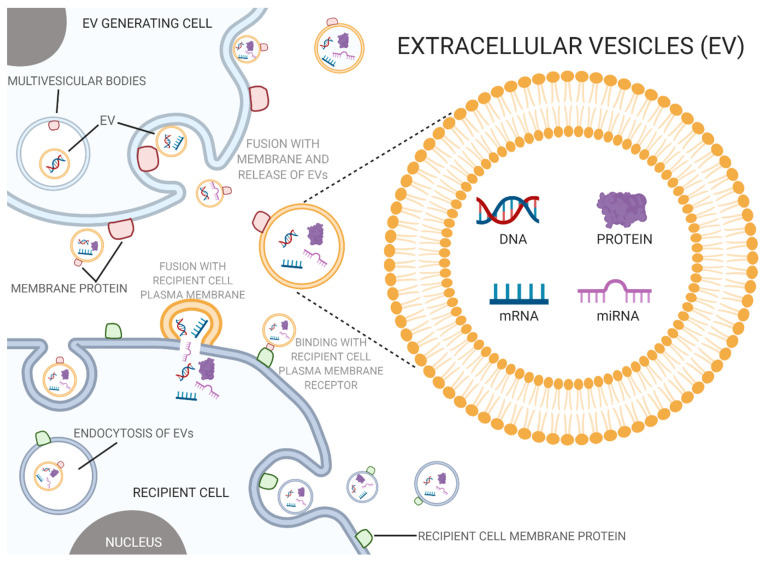
Extracellular vesicles and prenatal alcohol exposure (PAE). Extracellular vesicles act as a mode of intercellular communication, where PAE can alter the composition and content of these vesicles which in turn has been known to disturb normal biological processes and potentially contribute to PAE’s toxic and teratogenic effects (created with BioRender.com; accessed on 7 July 2021).

**Figure 2 ijms-22-08785-f002:**
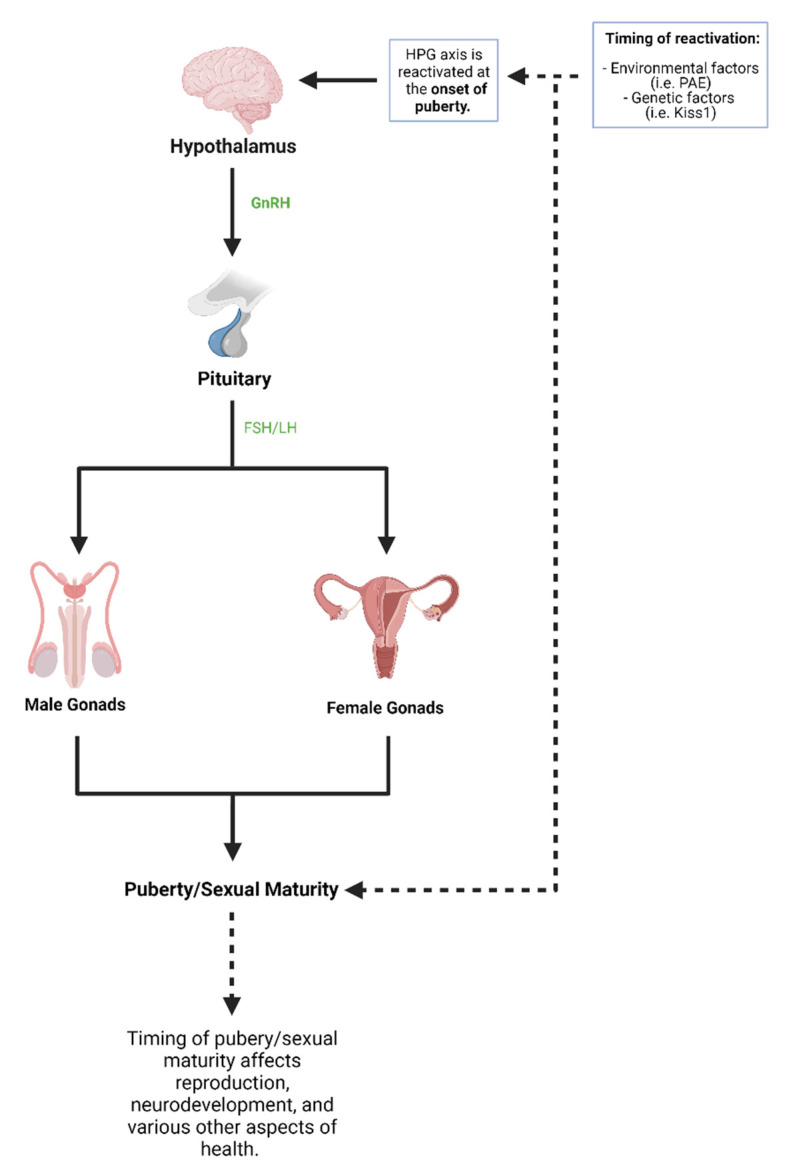
Puberty and hypothalamic-pituitary-gonadal (HPG) axis. The HPG axis works using various organs and hormones that lead to puberty and sexual maturity. The HPG axis is first active at birth and is reactivated during the start of puberty. Both environmental factors and genetic factors have been shown to influence the HPG axis and therefore the onset of puberty (created with BioRender.com; accessed on 7 July 2021).

**Figure 3 ijms-22-08785-f003:**
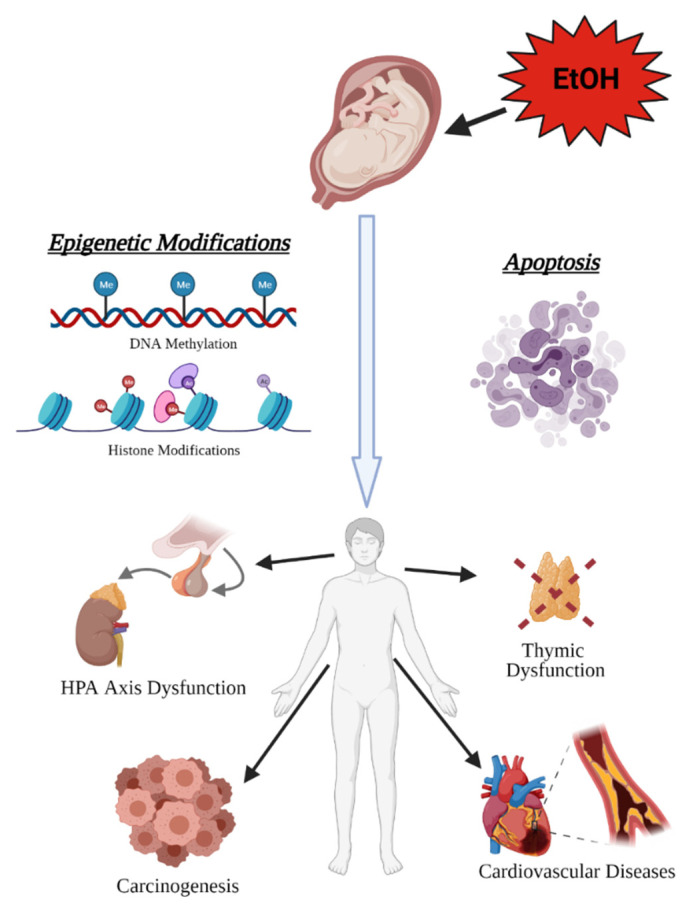
Prenatal alcohol exposure (PAE)has lifelong consequences on health. The toxic and teratogenic effects of alcohol exposure in utero manifest later in life as increased risk of hypothalamic–pituitary–adrenal (HPA) axis dysfunction, carcinogenesis, cardiovascular disease, and immune system dysfunction resulting from abnormal organogenesis (created with BioRender.com; accessed on 7 July 2021).

## Data Availability

Not applicable.

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
