# Peer review of "Toxic and Teratogenic Effects of Prenatal Alcohol Exposure on Fetal Development, Adolescence, and Adulthood"

_ijms, 2021, doi:10.3390/ijms22168785_

Round 1

Reviewer 1 Report

This is an insightful review of the literature on the topic of  the effects of prenatal alcohol on fetal development and long-lasting consequences of such exposure (through adolescence and adulthood). The literature on the topic is vast, thus the focus on particular aspects of PAE effects is completely justified.

However, the section on epigenetic mechanisms (announce in the Abstract as one of the main focuses of this Review) is disproportionally short. It could be expanded by addition of papers by Chen, Ozturk & Zhou (2013) (https://doi.org/10.1371/journal.pone.0060503) and  Dasmahapatru & Khan  (2015) (https://doi.org/10.1016/j.cbpc.2015.10.011). In addition, section on Cytokine and Immune Response could benefit from adding information about MB Heaton’s lab research (e.g.,  DOI: 10.1097/01.ALC.0000060527.55252.71 and DOI: 10.1016/s0165-3806(00)00032-8) as well as Raineki et al (Weinberg lab) work on cytokine and immune system development (DOI: 10.1016/j.bbi.2017.07.001 )

Section 3 is well-written, very informative and a fascinating read.

Author Response

Thank you for your thorough comments and helpful suggestions. We have revised our manuscript and expanded on the Epigenetic Mechanism section, including recommended studies by Chen, Ozturk & Zhou (2013) and Dasmahapatru & Khan (2015), and on the Cytokine and Immune Response section, including recommended studies by Heaton et al (2000, 2006) and Raineki et al (2017).

Reviewer 2 Report

The review paper by Chung et al. entitled “Toxic and Teratogenic Effects of Prenatal Alcohol Exposure on Fetal Development, Adolescence, and Adulthood” aims to discuss the literature regarding supporting epigenetic alterations as a mechanism contributing to an array of deficits associated with prenatal alcohol exposure. Overall, the authors provide a convincing analysis of the literature that strongly suggests that alterations in epigenetic modifications may result in a cascade of long-term effects that are dependent on amount and timing of ethanol exposure, in addition to the biological sex of the offspring. However, there were several concerns and questions that the authors should address to improve the impact and interpretations of this literature review.

  • Throughout the manuscript, for examples of studies that showed ethanol-induced effects, please provide the concentration/dose of ethanol exposure as well as the timing and duration of exposure. Perhaps if there is a common pattern, this can also be discussed.
  • Regarding miRNAs as a mechanism for PAE-induced alterations, given that in humans circulating miRNAs can be used as biomarkers, are there studies that have replicated the use of miRNAs as biomarkers in animal models? If so, are these consistent with the human work?
  • While the figures are nice and give a representative view of the various points that are discussed, there is no reference in the text to the figures. This would be helpful to the reader to know why the figures are there in the first place.
  • There is a typo at the end of line 297.
  • Epigenetic links to several of the subtopics (i.e. cardiovascular disease, congenital heart disease, cancer, etc) are mentioned throughout the manuscript and it is suggested that PAE may contribute to these changes. However, are the known PAE-induced epigenetic changes the same ones that have been identified in cardiovascular disease, congenital heart disease, cancer, etc? Perhaps providing some detail about the epigenetic contributions to these diseases, such as specific methylation and acetylation sites, or changes in epigenetic modification enzymes (DNMTS, HATs, HDACs) may better facilitate this potential link and the conclusions that are derived.
  • Although the information regarding miRNAs and EVs is important, it seems to be forgotten in section 3 and even in the conclusion section it is not directly tied back to the general thesis of the review paper.

Author Response

Thank you for your detailed suggestions. We have used your comments to revise and improve our manuscript. We have provided details of ethanol exposure (concentration, timing, and duration) throughout the manuscript, the use of miRNAs as biomarkers in animal models consistent with human subjects (Momen-Heravi et al., 2015), figure references in the text, comparison of epigenetic changes by PAE to other diseases, and connecting sentences that allow different sections to be more congruent with the general thesis of the manuscript.  

Throughout the manuscript, for examples of studies that showed ethanol-induced effects, please provide the concentration/dose of ethanol exposure as well as the timing and duration of exposure. Perhaps if there is a common pattern, this can also be discussed.

We have added details of ethanol exposure (concentration, timing, and duration) throughout the revised manuscript.

Regarding miRNAs as a mechanism for PAE-induced alterations, given that in humans circulating miRNAs can be used as biomarkers, are there studies that have replicated the use of miRNAs as biomarkers in animal models? If so, are these consistent with the human work?

We have added an alcohol study looking at the use of miRNAs as biomarkers in animal models consistent with human subjects (Momen-Heravi et al., 2015).

While the figures are nice and give a representative view of the various points that are discussed, there is no reference in the text to the figures. This would be helpful to the reader to know why the figures are there in the first place.

Figure references are added to the manuscript text.

There is a typo at the end of line 297.

This error has been corrected.

Epigenetic links to several of the subtopics (i.e. cardiovascular disease, congenital heart disease, cancer, etc) are mentioned throughout the manuscript and it is suggested that PAE may contribute to these changes. However, are the known PAE-induced epigenetic changes the same ones that have been identified in cardiovascular disease, congenital heart disease, cancer, etc? Perhaps providing some detail about the epigenetic contributions to these diseases, such as specific methylation and acetylation sites, or changes in epigenetic modification enzymes (DNMTS, HATs, HDACs) may better facilitate this potential link and the conclusions that are derived.

We have added how ethanol exposure changes epigenetic modification enzymes (section 2.1. Epigenetic Mechanisms). In addition, we have added the paradoxical mix of hyper- and hypomethylation that are seen both in PAE and cancer (subsection 3.2.1. Cancer) and acetylation of H3 histone lysine 9 as a link between PAE and congenital heart disease (subsection 3.2.4. Congenital Heart Disease).

Although the information regarding miRNAs and EVs is important, it seems to be forgotten in section 3 and even in the conclusion section it is not directly tied back to the general thesis of the review paper.

We have added connecting sentences to allow different sections to be more congruent with the general thesis of the manuscript, while still being able to stand alone and not dilute their specific points.